# Update on International Medical Taxonomies of Biomarkers and Their Applications in Management of Thyroid Cancers

**DOI:** 10.3390/diagnostics12030662

**Published:** 2022-03-09

**Authors:** Maria Trovato

**Affiliations:** Department of Clinical and Experimental Medicine, Policlinico Universitario, Consolare Valeria 1, 98125 Messina, Italy; mariatrovato@tin.it

**Keywords:** biomarkers taxonomies, thyroid cancers biomarkers, thyroglobulin, BRAF and RET/PTC molecular alterations, c-met expressions

## Abstract

Biomarkers (BMs) are medical signs which can be precisely measured and reproduced. Mainly, BMs provide information on the likely disease which can occur in an individual. On the other hand, BMs also signal disease recurrence in patients receiving therapy. The U.S. Food and Drug Administration coupled with the National Institutes of Health and the European Medicines Agency have proposed two distinct procedures to validate BMs. These agencies have elaborated two glossaries to describe the role of BMs. The aim of this study was to investigate medical taxonomies adopted by different governmental agencies for BM validation. Additional goals were to analyze efficiencies of the validated and candidate BMs for thyroid cancers (TCs). Currently, thyroglobulin is validated for monitoring TCs. Sorafenib-tosylate, Doxorubicin-hydrochloride, Vandetanib, Cabozantinib-s-malate, Dabrafenib-mesylate, Trametinib-dimethyl-sulfoxide, Lenvatinib-mesylate, Pralsetinib and Selpercatinib are validated for TC treatment. Among candidate BMs for TC diagnosis, there are molecular combinations including BRAF, RAS, RET/PTC and PAX8-PPARγ mutations. Noteworthy are BRAF and RET/PTC alterations already validated as targets of Dabrafenib-mesylate, Pralsetinib and Selpercatinib. Finally, cellular expressions of c-met in nodal TC metastases have diagnostic imaging applications. On the basis of this analysis, BM taxonomies should have common standards internationally recognized. BMs show different efficiencies depending on their diagnostic or therapeutic use.

## 1. Introduction

Biomarkers (BMs) are biological signs identified through two characters [1]. Firstly, each single BM is determined by a unit of measure [1]. Secondly, BMs have to be reproducible [1]. In 2015, BM definition was revised because the medical categories of BMs had inordinately grown [2]. By the U.S. Food and Drug Administration and the National Institutes of Health (FDA-NIH Biomarker Working Group), medical uses of this lexeme were clarified through a “living” glossary, namely, “Biomarkers, EndpointS, and other Tools” (BEST) Resource [2]. At large, BMs were defined as “indicators” either of physiological or pathological processes occurring in the biologic sphere [2,3]. As a consequence, medical BMs can be individualized through the examination of physiologic aspects, histologic features, radiographic traits and molecular qualities.

Currently, the BM catalogue is considered as strongly strategic for translational biomedicine aimed at finding future technologies which benefit human health [4,5]. In particular, several protein molecules have been proposed and validated as useful BMs to approach and cure cancerous diseases (see Table 1 in ref. [6]) [6]. A critical point for emerging cancer BMs is validation to confirm their own reliability [1,4,7,8]. On the basis of BEST Resource, BM validation consists of a multistep process by which to establish the performance of a BM as acceptable for its intended purpose [2,4]. Especially, the BM validation process includes both analytical and clinical validation steps [2].

In Western countries, different government agencies can issue the “Certificate of validation” for novel BMs [7]. For instance, in the USA the reliability of BMs is assured by fulfilling stringent criteria included in FDA Guidance [7,9,10]. In Europe, the performance of valid BMs has to exceed all the standards reported by the European Medicines Agency (EMA) [7,9,11].

The taxonomy of cancer BMs needs to be urgently regulated to enable it to be discussed in a unique global medical language [4,9,12]. This is due to the close relationship that exists between BMs and their use to diagnose and cure cancers [13]. BM taxonomy is essential as it provides details for comprehensive classifications and sufficient guidelines to evaluate diagnosis, prognosis and cancer risk combined [6].

The thyroid gland is constituted of two different types of epithelial cells. namely, follicular and parafollicular [14]. Both epithelial types are able to proliferate by causing thyroid cancers (TCs) [14]. Briefly, from follicular cells, follicular, papillary (PTC) and anaplastic thyroid carcinomas (ATC) arise, whereas medullary thyroid carcinomas come from parafollicular cells. To date, histological classification of TC variants remains the best BM for diagnosis and prognosis of thyroid malignancies [15,16]. However, numerous molecules have been proposed as adequate diagnostic and risk BMs for TCs in recent times [17,18,19].

This essay provides an analytic examination of the taxonomies of medical BMs that are currently adopted by different governmental agencies. The different types of medical language used to validate BMs by medical organizations in the USA and Europe was investigated.

There is a double aim: firstly, to document the BMs validated for TCs; secondly, to identify the possibilities of using BMs of TCs as diagnostic or therapeutic agents. To commence, the current use of qualified BMs is analysed. Next, the list of thyroid BMs adopted to cure TCs is described.

By exploring the diagnostic and therapeutic application of validated BMs, this investigation has especially displayed the different efficiencies of thyroid BMs. This review is concluded by assessing the clinical appropriateness of BMs used for TC diagnosis and therapy.

## 2. International Glossaries for BM Validation

In the last updated glossary of the BEST Resource in 2021, seven groups are used to compose the taxonomy of medical BMs (Table 1) [2,10]. Therefore, in the USA, BMs are archived under subcategories, i.e., susceptibility/risk, diagnostic, prognostic, monitoring, predictive, pharmacodynamic/response and safety, respectively (Table 1) [2,10] The BEST Resource taxonomy takes into account that a singular disease induces specific biological effects. Thence, this is principally an objective schematic subdivision of medical BMs. Further, this classification is associated with implementation of efficiency in diagnosing diseases and the development of precision therapies [9]. Conversely, the FDA taxonomy does not take into account the subjective data referring to a single patient, because specifically it “is not a measure of how an individual feels, functions, or survives” [2]. Indeed, a well-defined category of measure, namely, “clinical outcome assessment” (COA), is annotated in FDA premises [10]. Substantially, this note is associated with the above subjective qualifications. By monitoring clinical symptoms and the mental state of patients, COA is directly linked to outcomes of diseases in individual patients [10]. Currently, a restricted number of compounds have been ratified from the FDA as BMs useful in the clinical approach to cancers (see Table 1 in ref. [6]) [6,20].

In Europe, the EMA reports in its BM glossary “a biological molecule found in blood, other body fluids, or tissues that can be used to follow body processes and diseases in humans and animals” [11]. Seven subcategories of medical BMs are listed in the EMA glossary, similar to the BEST Resource taxonomy (Table 1) [7,9,11]. However, important differences between the two taxonomies are reported. In the EMA glossary, the medical BMs are registered overall: diagnostic, prognostic, predictive, enrichment, pharmacodynamic, safety signal and surrogate end point subcategories, respectively (Table 1) [7,9,11]. Ergo, the BEST Resource and the EMA classifications share only five subcategories (Table 1) [2,3,7,9,11]. The BEST Resource subcategories susceptibility/risk and monitoring BMs have no counterparts in the EMA taxonomy (Table 1).

Surrogate end point BMs deserve a separate discussion because the FDA taxonomy does not classify them under a specific subcategory of medical BMs (Table 1). Alternatively, the FDA distinguishes the surrogate end point subcategory from a true BM because it does not measure clinical benefit; it rather “predicts” clinical benefits on the basis of epidemiologic, therapeutic, pathophysiologic, or other scientific data [2]. In line with these data, the medical BM validation, established through the EMA certification, refers overall to “its use in pharmaceutical research and development”. In this context, EMA plays its part by publishing “opinions on the qualification of innovative development methods” [11].

Lastly, on the Asian continent, two government agencies have relevant competences relating to the program of BM qualification. One is based in China and the other in Japan [7]. The National Medical Products Administration, formerly the China Food and Drug Administration, is the competent authority in China [21], whereas, the Nipponese department responsible for BM nomenclature is the Japanese Pharmaceuticals and Medical Device Agency [22].

## 3. Epidemiological Monitoring, Valid BMs and New Proposals for TCs

TCs are common malignancies involving different ethnic groups [23]. According to the latest Annual Report to the Nation on the Status of Cancer epidemiological data available for 2020, in the USA incidence rates of TCs per 100,000 individuals were stable among males and females from 2012 through 2016 [24]. Henlei et al. have reported that average annual percentage change (AAPC), age-standardized, is not statistically significantly different from zero (stable) for TCs [24]. These data have revealed that AAPC is stably fixed at 0.5 in males; conversely, in females there is a decrement amounting to 0.1 percentage (see Figure 3 in ref. [24]) [24]. Biographical data and environmental exposure are indicated as signs of epidemiological significance for TCs. In fact, they have impinged upon the development of new cases for more than 50 years [25,26]. Notably, TCs carry a worse prognosis for older women [27,28]. The highest TC incidence rates have been noted among white adolescents and young adults [24]. In this group, TC risk factors also include low dose radiation exposure and excess body weight [24]. High TC diffusion has been topographically mapped at Chernobyl’s latitudes after the nuclear accident there [29]. Further, a major incidence of TCs has been reported in the population living in areas adjacent to the Etna volcano [30].

In the list of BM proteins for cancers of the FDA, thyroglobulin (Tg) only appears as qualified for TCs (see Table 1 in ref. [6]). Tg has been validated for protein tumor monitoring to use specifically in the clinical management of TCs [6,31]. Mainly, Tg certification for TCs dates back to 1997. By immunoassays on serum or plasma of patients, Tg levels have to be used to keep track of TCs derived from follicular cells [6,31]. Nevertheless, during the last 25 years several specific technical problems have developed when using Tg levels in the context of estimation of risk of recurrence (RR) for TCs. This is due to difficulties in calculating exactly the Tg cut-off point. Tg levels greater than 5 ng/mL are considered widely as alert signals for local recurrence or distant metastasis at six weeks from thyroidectomy. This is for subjects under thyroid hormone therapy [32]. In summary, the above data suggest that BM certifications should be periodically reviewed and monitored in the light of their effective and efficient use and then to evaluate whether they should be updated.

Between 2006 to 2020, nine molecules were approved by the FDA to use for TC treatment (Table 2) [33]. Currently, Sorafenib-tosylate, Doxorubicin-hydrochloride, Vandetanib, Cabozantinib-s-malate, Dabrafenib-mesylate, Trametinib-dimethyl-sulfoxide, Lenvatinib-mesylate, Pralsetinib and Selpercatinib have been validated to be used as drugs targeting molecules. This is because of their ability to recognize specific molecules expressed on the cancerous tissues of patients [33,34,35]. By targeting serine/threonine and tyrosine kinases such as RAF1, BRAF, VEGFR 1, 2, 3, PDGFR, KIT, FLT3, FGFR1 and RET, these compounds inhibit tumor cell proliferation and angiogenesis (Table 2) [33,34,35].

Since 2015, the American Thyroid Association (ATA) guidelines recommend the use of molecular combinations on cytological samples to clarify doubtful TC cases [36,37]. Basically, the ATA guidelines suggest the analysis of molecular mutations occurring in BRAF, RAS, RET/PTC and PAX8-PPARγ genes. In fact, there is scientific consensus supporting that, in preoperative indeterminate thyroid fine needle aspiration (FNA) samples, this molecular panel can significantly increase the diagnostic accuracy of TCs.

At the moment, four molecular tests are commercially available in the USA to diagnose and evaluate RR for TCs. Principally, each single test is able to detect BRAF as well as molecular alterations of RET on FNA samples [18,34]. Despite this, partial information is available on the effectiveness of molecular tests for cytopathology diagnosis of TCs. This is because molecular tests show high specificity and moderate sensitivity [17,19]. Along the same lines, there are preliminary data on RR evaluation of TCs through molecular tests. This is due to the absence of standard longer-term follow-up studies.

Several molecules have been correlated with the occurrence of TCs on histopathological samples [15,34]. The expressions of c-met, HGF, P53 isoforms and IL-6 in PTC appear as suitable BMs to recognize cancerous cells [38,39,40,41].

Particularly, c-met expression sets a paradigmatic example on how to use ex vivo investigations for clinical trial. In fact, in ex vivo lymph nodes, c-met expressions have been previously reported on the membrane of PTC metastatic cells [42]. Nearly 20 years later, these results are being investigated for in vivo diagnostic imaging applications. A clinical trial of phase I has been conducted in this regard [43]. The aim of this study was to test the usefulness of EMI-137 targeting c-Met for intraoperative imaging of PTC and nodal metastases (Table 3). By focusing on c-met expressions, a feasible approach to the detection of PTC nodal metastases has been developed by molecular fluorescent guided imaging.

Currently, Raman spectroscopic profiles are proving to be extremely reliable in identifying benign thyroid nodules [44].

## 4. Conclusions

In this study medical taxonomies used for regulatory acceptance of BMs have been examined.

Based on the evidence of this examination, it is clear that there is a deep need to build a common vocabulary globally. For example, by establishing common standards throughout the USA and Europe, BMs would be recognized in both countries. Additionally, an international taxonomy should work as a benchmark to record data on the effective and efficient use of BMs. In doing so, the performances of BMs would be evaluated and monitored by assessing unequivocal categories in different countries. This makes it possible to periodically review BMs’ efficiency. This demand for a unique taxonomy exists in order to record the biological levels of Tg in TCs. In fact, a large amount of data carefully documented is indispensable to be able to schedule an update of Tg status.

By analyzing validated BMs for TCs and the molecular panel recommended from ATA guidelines, several similarities and significant difference emerge (Table 2).

From a visual storytelling point of view, the molecular combinations used to improve TC diagnosis on cytological samples include BRAF and RET/PTC mutations. However, these molecular alterations have already been validated as BMs for treatment and management of advanced TCs. Therefore, from a practical perspective, FDA and EMA have validated the above molecular BMs exclusively for therapeutic use. This contradiction arises from the absence of an appropriate categorization aimed at quantifying the efficiency of molecular BMs in diagnostic and therapeutic fields.

Basically, molecular alterations of TC variants represent adequate therapeutic targets. Conversely, the same molecular defects are not completely suitable to recognize the presence of TC. For example, BRAF V600 mutation appears in about 40% of PTC and in 20–50% of ATC [45]. On this basis, Dabrafenib mesylate compound is included among molecular therapies targeting BRAF for TC cases that harbor BRAF V600 mutation (Table 2). In this regard, FDA has approved and included Dabrafenib mesylate as a TC drug (Table 2). In sharp contrast, the absence of BRAF mutation does not exclude the presence of PTC or ATC through molecular investigations. This is because BRAF mutation is insufficient to recognize all molecular types of PTC or ATC. That is exactly why both FDA and EMA have not validated BRAF V600 mutation as molecular BMs to use for diagnosis of TCs. Therefore, thyroid BMs display different efficiencies depending on whether the molecules are utilized as diagnostic or therapeutic targets. In common clinical practice, the efficiency of diagnostic molecular BMs is not interchangeable with that of therapeutic molecules used for TC management. This example clearly suggests that molecular BMs should have specific taxonomies, useful to assess the appropriateness of their medical use.

The best practices of molecular medicine for TC management include molecular therapies and recommend molecular diagnosis. At present, histological identification continues to be a satisfying indicator for diagnosis of TCs. Additional immunohistochemical and molecular expressions occurring in cancerous cells are more frequently regarded as diagnostic medical devices. In fact, their recognition plays relevant roles in TC diagnosis.

## Figures and Tables

**Table 1 diagnostics-12-00662-t001:** Taxonomy of medical BMs.

	BEST (FDA/NIH) Groups	EMA Groups
Biomarker subcategories	Susceptibility/risk	NA ^
	Diagnostic	Diagnostic
	Prognostic	Prognostic
	Monitoring	NA ^
	Predictive	Predictive
	NA ^	Enrichment
	Pharmacodynamic/response	Pharmacodynamic
	Safety	Safety signal
	Surrogate end point *	Surrogate end point *

^ NA: not applicable. * FDA taxonomy doesn’t classify surrogate end point BMs under a specific subcategory of medical BMs because they are predictors of clinical benefits. Data taken from references [2,10,11].

**Table 2 diagnostics-12-00662-t002:** Molecules approved for thyroid cancer therapy by the Food and Drug Administration.

Generic Names of Compound	Target Cancer	Class of Medications	Year First Posted
Sorafenib tosylate	Progressive, recurrent, or metastatic disease that does not respond to treatment with radioactive iodine	It blocks the enzyme RAF kinase, a critical component of the RAF/MEK/ERK signaling pathway, Further, it inhibits the VEGFR-2/PDGFR-beta signaling cascade	5 October 2006
Doxorubicin hydrochloride	Metastatic thyroid cancer	Anthracycline antibiotic	10 August 2007
Vandetanib	Medullary thyroid cancer	Tyrosine kinase inhibitors of vascular endothelial growth factor receptor 2 (VEGFR2)	29 April 2011
Cabozantinib-s-malate	Medullary thyroid cancer	Small molecule receptor tyrosine kinase (RTK) inhibitor *	19 December 2012
Dabrafenib mesylate	Anaplastic thyroid cancer	Inhibitor of B-RAF (BRAF)	21 June 2013
Trametinib dimethyl sulfoxide	Anaplastic thyroid cancer	Inhibitor of mitogen-activated protein kinase kinase (MAP2K; MAPK/ERK kinase; MEK) 1 and 2	21 June 2013
Lenvatinib mesylate	Progressive, recurrent, or metastatic disease that does not respond to treatment with radioactive iodine	Inhibitor of vascular endothelial growth factor receptor 2 (VEGFR2, also known as KDR/FLK-1) tyrosine kinase	26 February 2015
Pralsetinib	Medullary thyroid cancer, metastatic or advanced thyroid cancer RET fusion gene thyroid cancer	Inhibitor of mutant forms of and fusion products of proto-oncogene receptor tyrosine kinase RET	9 October 2020
Selpercatinib	Medullary thyroid cancer and thyroid cancer that has a RET fusion gene and is metastatic or advanced.	Kinase inhibitor of wild-type, mutant and fusion products involving the proto-oncogene receptor tyrosine kinase rearranged during transfection (RET)	26 May 2020

* Among these RTK inhibitors are included inhibitors of hepatocyte growth factor receptor (MET), RET (rearranged during transfection), vascular endothelial growth factor receptor types 1 (VEGFR-1), 2 (VEGFR-2), and 3 (VEGFR-3), mast/stem cell growth factor (KIT), FMS-like tyrosine kinase 3 (FLT-3), TIE-2 (TEK tyrosine kinase, endothelial), tropomyosin-related kinase B (TRKB) and AXL. Data taken from https://www.cancer.gov/about-cancer/treatment/drugs/thyroid (accessed on 19 January 2021).

**Table 3 diagnostics-12-00662-t003:** Summary of data posted for “Precision thyroid cancer surgery with molecular fluorescent guided imaging” clinical trial *.

Start Date	Thyroid Cancer	Primary Purpose	Drug Agent	Devices	Phase of Study	Completion Date
2018	Lymph Node Metastases of Papillary Thyroid Cancer	Diagnostic	EMI-137 **	Multispectral Fluorescence Reflectance Imaging (Spectroscopy)	I	2019

* ClinicalTrials.gov Identifier: NCT03470259. ** Fluorescence molecular imaging agent that targeting c-Met, peak emission at 675 nm range. Information provided by (Responsible Party): Schelto Kruijff, MD PhD, University Medical Center Groningen.

## Data Availability

Not applicable.

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
