# Peer review of "Update on International Medical Taxonomies of Biomarkers and Their Applications in Management of Thyroid Cancers"

_diagnostics, 2022, doi:10.3390/diagnostics12030662_

Round 1

Reviewer 1 Report

The article presents a review of the taxonomies of biomarkers for thyroid cancer. The style is clear and comprehensive, attractively presented, the data being synthesized from an original perspective.

However, we suggest that the authors re-evaluate some of the cited sources (are all the 8 self-citations really necessary and best references for this article? Some of them are too generally cited: line 198, citations 39-43). And other very recent references do not appear, for example:

Agarwal S, Bychkov A, Jung CK. Emerging Biomarkers in Thyroid Practice and Research. Cancers (Basel). 2021 Dec 31;14(1):204. doi: 10.3390/cancers14010204. PMID: 35008368; PMCID: PMC8744846.

We suggest checking the uniformity of writing the references (see 26, 27, 46 ...) .

A few small  other remarks, too:

- line 17: please replace “Diagnostic” wits “Diagnostics”.

- line 136: please replace “biomarker” with “biomarker’s”

- line 144: “._According” – seems to be a typo, please verify

- line 206: “spectroscopy for transitional biomedicine” – please verify, does not seem to be the correct name of the technique.

Author Response

Dear Reviewer,

Thank you for your kind revisions concerning my manuscript entitled “UPDATE ON INTERNATIONAL MEDICAL TAXONOMIES OF BIOMARKERS AND THEIR APPLICATIONS IN MANAGEMENT OF THYROID CANCERS” (Diagnostics-1610091).

Your comments are all valuable and very helpful for revising and improving the paper. I have made corrections which I hope can meet with your approval.

The article presents a review of the taxonomies of biomarkers for thyroid cancer. The style is clear and comprehensive, attractively presented, the data being synthesized from an original perspective.

I thank the reviewer for these comments!

However, we suggest that the authors re-evaluate some of the cited sources (are all the 8 self-citations really necessary and best references for this article? Some of them are too generally cited: line 198, citations 39-43).

Thanks for the reviewer’s suggestion. I have accepted the reviewer’s comments and deleted the citation 42.

And other very recent references do not appear, for example:

Agarwal S, Bychkov A, Jung CK. Emerging Biomarkers in Thyroid Practice and Research. Cancers (Basel). 2021 Dec 31;14(1):204. doi: 10.3390/cancers14010204. PMID: 35008368; PMCID: PMC8744846.

Thanks for the reviewer’s advice. It was my oversight; I sincerely accepted the reviewer’s suggestions and added reference (34).

We suggest checking the uniformity of writing the references (see 26, 27, 46 ...).

Thanks for the reviewer’s suggestion. I revised references.

A few small other remarks, too:

- line 17: please replace “Diagnostic” wits “Diagnostics”.

Thank you. I replaced “Diagnostic” with “Diagnostics” as the reviewer’s suggestion.

- line 136: please replace “biomarker” with “biomarker’s”

Thank you. I replaced “biomarker” with “BM’s” as the reviewer’s suggestion.

- line 144: “._According” – seems to be a typo, please verify

Thank you. I deleted _

- line 206: “spectroscopy for transitional biomedicine” – please verify, does not seem to be the correct name of the technique.

Thank you. I replaced “spectroscopy for transitional biomedicine” with “molecular fluorescent guided imaging” as the reviewer’s suggestion.

Yours sincerely,

Maria Trovato

Reviewer 2 Report

It's very interesting article. Correct presentation of the topic.
The process of validation of biomarkers was properly discussed.
Correct selection of references.

Author Response

Dear Reviewer,

Thank you for your kind revisions concerning my manuscript entitled “UPDATE ON INTERNATIONAL MEDICAL TAXONOMIES OF BIOMARKERS AND THEIR APPLICATIONS IN MANAGEMENT OF THYROID CANCERS” (Diagnostics-1610091).

Your comments are all valuable. I have made corrections which I hope can meet with your approval.

It's very interesting article.

I thank the reviewer for these comments!

Correct presentation of the topic.
The process of validation of biomarkers was properly discussed.
Correct selection of references. It's very interesting article. Correct presentation of the topic.
The process of validation of biomarkers was properly discussed.
Correct selection of references.

I thank the reviewer for this kind revision!

Yours sincerely,

Maria Trovato

This manuscript is a resubmission of an earlier submission. The following is a list of the peer review reports and author responses from that submission.

Round 1

Reviewer 1 Report

This is a well written paper. The study discusses the importance of biomarkers in classifications of thyroid cancers.

This is an important paper as this is very relevant to the literature. We do see incorporation of biomarkers in staging (AJCC staging 8th edition for breast cancer).

Introduction : Can you add any relevant pieces of previous work cited?

Also in the introduction , please make a coherent case as to why this research question is important.

The paper talks historically about the classification, but can strengthen the paper by identifying the range of challenges and opportunities and provide a framework for the validation ahead. ( some of it is mentioned 8 yr period), but a succinct description would be helpful.

Author Response

Response to Reviewer 1 Comments

Dear Reviewer,

thank you very much for your kind revisions, useful comments and constructive suggestions.

Please, by following your suggestions, Introduction and Conclusions sections have been revised; further, new references have been added.

Point 1:

Can you add any relevant pieces of previous work cited? Also in the introduction, please make a coherent case as to why this research question is important.

Response 1:

In introduction section has been added:

“Nonetheless, numerous molecules have been proposed as adequate diagnostic and risk BMs for TC [25-27]. Mainly, to diagnose and evaluate TC risk of recurrence (RR), molecular tests have been developed to applied on thyroid cytopathology. However, these tests are in an experimental phase because of high specificity and moderate sensitivity [26, 27]. Further, the ability of molecular tests to detect RR remains indeterminate because of standard longer-term follow-up studies are absent.  In fact, molecular tests associate with cytological diagnosis are just recommended from American thyroid association. On the other hand, any molecular test is included in TC classifications. Where molecular therapeutic targets are concerned, however, it is a different matter. This is due to FDA and EMA validations for use of molecular targets as drugs for treatments and managements of advanced TCs [28]

Point 2: The paper talks historically about the classification, but can strengthen the paper by identifying the range of challenges and opportunities and provide a framework for the validation ahead. ( some of it is mentioned 8 yr period), but a succinct description would be helpful.

Response 2:

In conclusions section has been added:

“Basically, molecular alterations of TC variants represent adequate therapeutic targets. Conversely, the same molecular defects are not completely suitable to recognize the presence of TC, in generally. For example, BRAF V600 mutation appears in about 40% of PTC and in 20-50% of ATC [54]. On these basis, Dabrafenib mesylate compound is included among molecular therapies targeting BRAF for TCs cases that harbor BRAF V600 mutation (Table 2). In sharp contrast, the absence of BRAF mutation doesn’t exclude the presence of PTC or ATC in molecular diagnostic field. This is because BRAF mutation only is insufficient to recognize all PTC or ATC molecular types. In this regarding, FDA has approved and included Dabrafenib mesylate as TC drug (Table 2). However, both FDA and EMA have not validated BRAF V600 mutation as molecular BMs for TCs. This example clearly shows that diagnostic molecular BMs have a different semiology in respect with therapeutic molecules and then, they are not interchangeable in the common clinical practice. 

Therefore, best practices of molecular medicine for TCs management include molecular therapies and recommend molecular diagnosis. In fact, histological identification continues to be a satisfying indicator to make diagnosis of TCs. At present, additional immunohistochemical and molecular recognitions have adjuvant roles for TCs diagnosis. To define TCs behaviour still remains the basic problem.

The average time for reviewing the international classifications for TCs is estimated to be more than seven years. This is a reasonable amount of time to monitor an adequate number of individuals and then, to carefully document histological and molecular features of TCs paired with clinical outcome of patients. This is because of longitudinal studies over long periods of time are the most appropriate tool to demonstrate the actual usefulness and necessity of TC classifications or guidelines, too. On this line, longer-term follow-up studies of new candidate molecular BMs have to be performed for RR assessment. Accurate evaluations of serum Tg levels are need for initial RR estimation [55]. This is because Tg cut-off point has not been yet exactly calculated. Mainly, Tg levels greater than 5 ng/mL are considered alert signals for local recurrence or distant metastasis at six weeks from thyroidectomy in subjects under thyroid hormone therapy”.

Reviewer 2 Report

In this paper the author presents a review on the use of biomarkers in thyroid cancer management.

This is an interesting paper and the topic of biomarkers in thyroid cancer is discussed in an interesting way however the paper has a very limited discussion of the biomarkers used in current thyroid cancer management such as thyroglobulin and those used in cytology and I am unsure as to how useful this review to those interested in thyroid cancer biology or management.

The language used in the paper is nonconventional and difficult for me to understand and I feel the paper would benefit from being edited by a writing service. Examples of this confusing language include ‘Standard histological BMs are currently used to memorize thyroid cancers (TCs) features for classifications’ which presumably means – histological biomarkers are an accepted way to classify different types of thyroid cancer.  

In the abstract assay presumably refers to essay?

Some of the references used do not support the claims made, an example being ref 4 which is a reference for a review which describes the process from biomarker discovery to translation rather than biomarkers as a cure for cancer. Also ref 10 is referring to thyroid cancer epidemiology but a paper on growth factors is referenced

The use of the weblinks in the paper are distracting to the reader and would be helpful if they were included with the references with the date of when the sites were accessed given many are likely to change with time.

While the paper is interesting much of the paper describes the history of organisations involved in cancer classification which to me seems to have limited relevance to the current topic and perhaps a more appropriate title for the paper would be ‘history of biomarkers and application to thyroid cancer management’

Author Response

Response to Reviewer 2 Comments

Dear Reviewer,

thank you very much for your kind revisions and useful comments.

Please, by following your suggestions: Introduction and Conclusions sections have been revised; further, new references have been added.

Point 1:

This is an interesting paper and the topic of biomarkers in thyroid cancer is discussed in an interesting way however the paper has a very limited discussion of the biomarkers used in current thyroid cancer management such as thyroglobulin and those used in cytology and I am unsure as to how useful this review to those interested in thyroid cancer biology or management.

Response 1:

In introduction section has been added:

“Nonetheless, numerous molecules have been proposed as adequate diagnostic and risk BMs for TC [25-27]. Mainly, to diagnose and evaluate TC risk of recurrence (RR), molecular tests have been developed to applied on thyroid cytopathology. However, these tests are in an experimental phase because of high specificity and moderate sensitivity [26, 27]. Further, the ability of molecular tests to detect RR remains indeterminate because of standard longer-term follow-up studies are absent.  In fact, molecular tests associate with cytological diagnosis are just recommended from American thyroid association. On the other hand, any molecular test is included in TC classifications. Where molecular therapeutic targets are concerned, however, it is a different matter. This is due to FDA and EMA validations for use of molecular targets as drugs for treatments and managements of advanced TCs [28]

In conclusions section has been added:

Basically, molecular alterations of TC variants represent adequate therapeutic targets. Conversely, the same molecular defects are not completely suitable to recognize the presence of TC, in generally. For example, BRAF V600 mutation appears in about 40% of PTC and in 20-50% of ATC [54]. On these basis, Dabrafenib mesylate compound is included among molecular therapies targeting BRAF for TCs cases that harbor BRAF V600 mutation (Table 2). In sharp contrast, the absence of BRAF mutation doesn’t exclude the presence of PTC or ATC in molecular diagnostic field. This is because BRAF mutation only is insufficient to recognize all PTC or ATC molecular types. In this regarding, FDA has approved and included Dabrafenib mesylate as TC drug (Table 2). However, both FDA and EMA have not validated BRAF V600 mutation as molecular BMs for TCs. This example clearly shows that diagnostic molecular BMs have a different semiology in respect with therapeutic molecules and then, they are not interchangeable in the common clinical practice. 

Therefore, best practices of molecular medicine for TCs management include molecular therapies and recommend molecular diagnosis. In fact, histological identification continues to be a satisfying indicator to make diagnosis of TCs. At present, additional immunohistochemical and molecular recognitions have adjuvant roles for TCs diagnosis. To define TCs behaviour still remains the basic problem.

The average time for reviewing the international classifications for TCs is estimated to be more than seven years. This is a reasonable amount of time to monitor an adequate number of individuals and then, to carefully document histological and molecular features of TCs paired with clinical outcome of patients. This is because of longitudinal studies over long periods of time are the most appropriate tool to demonstrate the actual usefulness and necessity of TC classifications or guidelines, too. On this line, longer-term follow-up studies of new candidate molecular BMs have to be performed for RR assessment. Accurate evaluations of serum Tg levels are need for initial RR estimation [55]. This is because Tg cut-off point has not been yet exactly calculated. Mainly, Tg levels greater than 5 ng/mL are considered alert signals for local recurrence or distant metastasis at six weeks from thyroidectomy in subjects under thyroid hormone therapy.

Point 2:

The language used in the paper is nonconventional and difficult for me to understand and I feel the paper would benefit from being edited by a writing service. Examples of this confusing language include ‘Standard histological BMs are currently used to memorize thyroid cancers (TCs) features for classifications’ which presumably means – histological biomarkers are an accepted way to classify different types of thyroid cancer. 

Response 2:

  1. editing service has been required.
  2. I agree with reviewer 2 to replace ‘Standard histological BMs are currently used to memorize thyroid cancers (TCs) features for classifications’ sentence with “histological biomarkers are an accepted way to classify different types of thyroid cancer”.

Point 3:

In the abstract assay presumably refers to essay?

Response 3:

It has been revised.

Point 4:

Some of the references used do not support the claims made, an example being ref 4 which is a reference for a review which describes the process from biomarker discovery to translation rather than biomarkers as a cure for cancer. Also ref 10 is referring to thyroid cancer epidemiology but a paper on growth factors is referenced

Response 4:

I don’t agree with Reviewer 2. This is because ref 4 (now this is ref 6) ([Füzéry AK, Levin J, Chan MM, Chan DW. Translation of proteomic biomarkers into FDA approved cancer diagnostics: issues and challenges. Clin Proteomics. 2013 Oct 2;10(1):13. doi: 10.1186/1559-0275-10-13. PMID: 24088261; PMCID: PMC3850675) includes a Table showing the a list of FDA-approved protein tumor markers currently used in clinical practice. This table is associated with specific comments.

In Ref 10 is included a paragraph referring to thyroid cancer epidemiology. However, a new reference has been added (Olson E, Wintheiser G, Wolfe KM, Droessler J, Silberstein PT. Epidemiology of Thyroid Cancer: A Review of the National Cancer Database, 2000-2013. Cureus. 2019 Feb 24;11(2):e4127. doi: 10.7759/cureus.4127.

Point 5:

The use of the weblinks in the paper are distracting to the reader and would be helpful if they were included with the references with the date of when the sites were accessed given many are likely to change with time.

Response 5:

References have been added.

Point 6:

While the paper is interesting much of the paper describes the history of organisations involved in cancer classification which to me seems to have limited relevance to the current topic and perhaps a more appropriate title for the paper would be ‘history of biomarkers and application to thyroid cancer management’

Response 6:

Title of paper has been changed in “SEMIOLOGY OF BIOMARKERS AND THEIR APPLICATION TO THYROID CANCER CLASSIFICATIONS”.

Reviewer 3 Report

I make no comment. The paper adds new cognitive elements. It is well written.

Author Response

Dear Reviewer,

thank you very much for your remarkable appreciation.